# Lower Extremity Amputation and Peripheral Revascularisation Rates in Romania and Their Relationship with Comorbidities and Vascular Care

**DOI:** 10.3390/jcm13010052

**Published:** 2023-12-21

**Authors:** Stefan Ionac, Steven K. Rogers, Cosmina I. Bondor, Frank L. Bowling, Iulia Iovanca Dragoi, Mihai Ionac

**Affiliations:** 1CerVasc, Vascular and Endovascular Surgery Research Center, Faculty of Medicine, Victor Babes University of Medicine and Pharmacy, Piata Eftimie Murgu 2, 300041 Timisoara, Romania; iulia.dragoi@umft.ro; 2School of Medical Sciences, Faculty of Biology, Medicine and Health, University of Manchester, Manchester Academic Health Science Centre, Manchester University NHS Foundation Trust, Oxford Road, Manchester M13 9WL, UK; 3Manchester Academic Vascular Research and Innovation Centre (MAVRIC), Manchester University NHS Foundation Trust, Oxford Road, Manchester M13 9WL, UK; 4Department of Medical Informatics and Biostatistics, Iuliu Hațeganu University of Medicine and Pharmacy, 6 Pasteur Street, 400349 Cluj-Napoca, Romania; 5Clinic of Vascular Surgery, Victor Babes University of Medicine and Pharmacy, Piata Eftimie Murgu 2, 300041 Timisoara, Romania; mihai.ionac@umft.com

**Keywords:** major amputation, minor amputation, revascularisation, Romania, peripheral arterial disease, diabetes mellitus

## Abstract

(1) Background: This retrospective Romanian study aimed to calculate the rate of, and comparison between, amputation and revascularisation for patients with either cardiovascular or diabetic comorbidities. (2) Materials: In our hospital-based database, we analysed patient-level data from a series of 61 hospitals for 2019, which covers 44.9% of the amputation patients for that year. The national database is compiled by the national houses of insurance and was used to follow amputations and revascularisations between 2016 and 2021. (3) Results: During the six-year period, the mean number of amputations and revascularisations was 72.4 per 100,000 inhabitants per year for both groups. In this period, a decline in open-surgical revascularisation was observed from 58.3% to 47.5% in all interventions but was not statistically significant (r = −0.20, *p* = 0.70). The mean age of patients with amputation (hospital-based database) was 67 years. Of these patients, only 5.1% underwent revascularisation in the same hospital prior to amputation. The most common comorbidities in those undergoing amputations were peripheral arterial disease (76.8%), diabetes (60.8%), and arterial hypertension (53.5%). Most amputations were undertaken by general surgeons (73.0%) and only a small number of patients were treated by vascular surgeons (17.4%). (4) Conclusions: The signal from our data indicates that Romanian patients probably have a high risk of amputation > 5 years earlier than Western European countries, such as Denmark, Finland, and Germany. The prevalence of revascularisations in Romania is 64% lower than in the Western European countries.

## 1. Introduction

Peripheral arterial disease (PAD) affects 10–20% of people >60 years old [1], and 22–37% of these people progress to a critical point where the limb is threatened, which sometimes results in amputation if not promptly revascularised. Importantly, it was reported in 1997 by Danish academics that as few as 30% of patients are assessed by a vascular surgeon prior to an amputation [2]. Moreover, since 2005, as many as 50% of amputations worldwide have been reported to occur with no prior vascular diagnosis [3], even though 64.4% of those who receive a major amputation will be deceased after 5 years [4]. Although Western Europe has significantly improved in this regard, Romania has fallen behind, with the aforementioned figures still being observed today.

In 2002, vascular surgery was officially recognised as an independent specialty in Romania, but as important as it may be, it does not exist as a compulsory lecture in the undergraduate medical curriculum of Romanian universities. Since 2017, the vascular surgery residency curriculum has been harmonised with European recommendations.

At present, the number of Romanian citizens is slightly above nineteen million [5]. Currently, 18% of the population is >64 years old [6]. Therefore, a growing elderly population is expected, with more than 30% expected to be >65 years old by 2050, making Romania the country with the 25th largest aging population in the world.

The most important comorbidities in those with PAD are arterial hypertension, diabetes, and smoking [7]. Blood pressure (BP) higher than 160/95 mmHg is associated with a 2.5-fold and 4-fold greater risk of developing PAD in men and women, respectively [8]. The prevalence of arterial hypertension in the Romanian adult population is 11% [9].

Patients with diabetes mellitus (DM) have a 2- to 4-fold higher risk of developing PAD, where adequate glycaemic control is highly beneficial, as every unit rise in HbA1c increases the risk of developing PAD by 28% [10]. When considered that 11.6% of the adult population has diabetes, it makes PAD a critical comorbidity that doctors and patients must address [11]. A European study has already demonstrated that in Western Europe, the prevalence of major lower extremity amputations (LEA) is 20 per 100,000 people whereas in Eastern Europe it is above 30 [12]. One possible explanation could be the difference in revascularisation efforts and the difference in smoking prevalence between Eastern and Western Europe [13,14].

To assess the current rate of revascularisation and amputation procedures, a retrospective nationwide data analysis of procedures and outcomes submitted to Romanian national authorities was conducted, focusing on the number of amputations and revascularisations with vascular and diabetic comorbidities, as well as the vascular care that Romanian patients receive. By calculating the rates of incidence and prevalence across Romania, this research aims to inform decision makers to improve amputation rates.

## 2. Materials and Methods

### 2.1. Design

All hospital data were submitted to the Romanian National Health Insurance House (NHIH) for hospitals outside of Bucharest and to the National School of Public Health, Management and Professional Development, Bucharest (NSPHMDPB).

There are 3 hospital levels (regional, county, and local) where most are university-affiliated [15].

This study utilised population-based data collected by NHIH and NSPHMDPB between 2016 and 2021 to create a singular national-level research database. A patient-level anonymised data set was then created, and subsequently, data were extracted from a group of hospitals for the year 2019 and were shared with the study team.

The REporting of studies Conducted using Routinely collected health Data (RECORD) statement and Strengthening the Reporting of Observational Studies in Epidemiology (STROBE) guidelines were followed [16].

Since anonymised data were used, no patient informed consent or ethics committee approval was required.

### 2.2. Setting/Sources

**Hospital-based database (HBD):** In 2019, 227/532 public and private hospitals [15] reported at least one amputation and/or revascularisation procedure. Hospitals that performed < 15 amputations were removed from the analysis, leaving 145 (63.9%) hospitals which covered 95.9% of all procedures. Forty-one (28.3%) of them were university hospitals, 29 were (20%) county hospitals, and 75 (51.7%) were municipal, city, or private hospitals. From the 145 hospitals, a core series of 61 provided access to patient data to create a group of hospitals as representative as possible. This group represented 42.1% of valid hospitals from the national database in 2019 which covered ~50% of the amputations reported (44.9% first procedures and 44.7% secondary procedures) in 2019 (Table 1). From this list of hospitals, patients with primary and secondary procedures of amputations and/or revascularisations were extracted. Additionally, primary and/or secondary diagnosis of PAD, alcohol consumption, DM, smoking, hyperlipidaemia, HTA, obesity, ischemic stroke, coronary disease, and carotid artery disease were collected. Specific disease and procedural codes are included in Appendix A.

#### 2.2.1. Inclusion Criteria for the Hospital Database

Cases that had at least one primary or a secondary reported amputation and/or revascularisation were included in this study.

#### 2.2.2. Exclusion Criteria for the Hospital Database

The same exclusion criteria as above were used. Five hundred and three (7.75%) cases with a first diagnosis of trauma, procedure doubling, patients under 18 years, and patients with burns were excluded.

**The National database (ND)** includes the national population of state-admitted patients who visited the hospital between the 1st of January 2016 to the 31st of December 2021. Data were obtained from the NHIH and NSPHMPDB. Each patient has a main disease/procedure assigned, which is the primary disease/procedure with which they are admitted. Everything else is considered a secondary disease/procedure. All the patients’ demographics, amputations, and outcomes were extracted. Hospitals that reported at least one case of amputation and/or revascularisation as the first procedure were counted.

If readmitted within one year, the patient was counted only once by using a unique anonymous case identification (ID) number to identify multiple readmissions.

#### 2.2.3. Exclusion Criteria for the Nationwide Database

Cases with a first diagnosis of trauma, burns, and malignancies, as well as cases with patients under 18 years and with procedure doublings (i.e., the same first and secondary procedures were reported), have been excluded. A total of 6547 cases were excluded (Figure 1).

**Data from the questionnaire:** In 2022, the Romanian Society of Vascular Surgery conducted a questionnaire where they asked every hospital about vascular surgery facilities. With the information extracted from the results, the map in Figure 2B was developed to compare the locations of vascular surgery centres and where amputations occurred.

#### 2.2.4. Inclusion Criteria for the Questionnaire

Vascular centres were defined as having a minimum of 25 beds, access to the operating room, and diagnostic imaging facilities. Smaller vascular centres, called compartments, were defined as having a minimum of 10 beds.

#### 2.2.5. Definitions

Major amputations were those above the ankle joint [17]. An episode of hospitalisation was the interval between admission and discharge, regardless of possible readmissions.

### 2.3. Variables

#### 2.3.1. National Database

The number and type of amputation (Appendix A) and revascularisation (Appendix A) performed between 2016–2021 were collated.

#### 2.3.2. Hospital-Based Database

Risk factors (Appendix A), discharge clinic (Appendix A), operating surgeon’s specialty (Appendix A), type of surgical procedure (Appendix A), amputation level (Appendix A), number of amputations with no prior revascularisation in the same hospital episode, in-hospital-death, and demographic variables (age and county of residence) were collected between 2016–2021.

#### 2.3.3. Statistical Analysis

By using the hospital-based database, the number of patients excluded was identified (43.4% of the patients in the year 2019). Thus, for the rest of the years, the same exclusion percentages for primary (malignancies, poison, and traumas—1.6%; burns—0.1%; under 18 years old—0.4%) and secondary (1.7%, 0.3%, and 0.2%) amputations were used for the national database.

All the valid hospitals were considered, and their population was used to select hospitals for the Hospital-Based Database (HBD). The representativity of this sample was evaluated from the distribution of hospital types, the distribution of amputation frequencies on procedure types, and the distribution of revascularisation frequencies on procedure types by using the test for one proportion.

The estimation of the cases of amputations that could be excluded from the National Database (ND) each year was calculated based on the obtained frequencies of the excluded cases. The proportion of excluded cases from the total number of reports on codes related to amputation was computed in the HBD. This percentage was also subtracted from the number of reported cases each year obtained from the ND. The result was the estimation of reported cases of amputation per year in all hospitals without cases that were reported as a secondary procedure in those under 18 years, due to procedure doublings, or due to trauma, burns, or malignancies.

The rate was computed in reference to the total resident population by the first of January of the respective year. Standardisation was made per 10^5^ population.

Trends in the considered time period for the number of amputations/revascularisations were evaluated with Pearson correlation for a linear trend and with Spearman correlation for nonlinear trends. Maps were made with Microsoft Excel (v16.7).

Odds ratios were computed using a control group of the cases reported in 2019 at the same hospitals. Relative risk (RR) for mortality was computed in comparison with the category of those under 30 years old. Then, 95% confidence intervals were computed for frequencies, RR, and OR. The level of error was chosen to be α = 0.05, and statistical analysis was performed using SPSS 25.0, MedCalc (v. 22.016), and Microsoft Excel (v16.7).

## 3. Results

### 3.1. The National Database

A decline in open-surgical revascularisation was observed (Figure 3, r = −0.20, *p* = 0.70). However, before the SarsCOV-2 pandemic, data showed an increasing tendency toward endovascular techniques (r = 0.63, *p* = 0.37), which represented 51.2% of all interventions. Balloon angioplasties with a single stent were the most popular (25.4%). Bypasses and patch angioplasties accounted for <30% of all the revascularisations performed, while others were balloon angioplasties with multiple stents (14.4%), embolectomies (22.7%) and others (Appendix A). Revascularisations increased during the observation period, but this was not statistically significant (r = 0.47, *p* = 0.348). However, the increase was statistically significant in pre-pandemic years (r = 0.98, *p* = 0.018). The same trends were observed for the number of endovascular procedures (5324 in 2016 to 8484 in 2021), which increased over time (r = 0.74, *p* = 0.092) but was statistically significant during pre-pandemic years (r = 0.97, *p* = 0.027). The detailed data about the types of intervention are displayed in Appendix A.

Across the timeframe, the mean number of amputations and revascularisations was 66.8/100,000/year and revascularisations were 72.4/100,000 (Figure 4), the most prevalent being above the knee (29.9%). Major amputations represent 39.5% of all the amputations. The second most frequent procedure was toe amputation at a rate of 26.3% (details in Appendix A). During the pre-pandemic years, there was a slight increase in the number of minor amputations (r = 0.90, *p* = 0.11). Minor and major amputations maintained a steady frequency pre- and peri-pandemic (Figure 3, r = 0.40, *p* = 0.60). Total numbers of amputations and revascularisations are provided in Appendix A.

### 3.2. Testing the Representability of the National Database Using the Hospital-Based Database

From the distribution of the type of hospitals, the 61 selected hospitals in the national database were a representative sample for the year 2019 (*p* ≥ 0.640). When comparing the proportion of each code related to amputation and revascularisation with the corresponding proportion in the population, we found the 61 pool hospitals representative of the entire country. However, this was not the case for all codes when considering revascularisation. The sample proportions were not significantly different (Appendix A). Some proportions were significantly different than the proportion reported by the hospitals to the national database when considering revascularisation (Appendix A).

### 3.3. The Hospital-Based Database

#### 3.3.1. Amputations/Medical Comorbidities

The mean age of patients who received an amputation was 67 years and 38.4% were <65 years. Five percent of patients who received secondary amputation underwent primary revascularisation in the same admission year. The most common comorbidities were PAD (76.8%), diabetes (60.8%), and hypertension (53.5%). A total of 7.3% of patients who received amputation smoked. Detailed comorbidity analysis is included in Table 2 and Appendix A.

#### 3.3.2. Administrative Risk Factors

More than 73% of amputations were performed by general surgeons and 17.4% by vascular surgeons (Appendix A), with the same proportions observed for the type of ward at discharge (Appendix A).

#### 3.3.3. Age-Related Risk Factors

Patients who were 55 to 85 years old represented 80.6% of all amputation cases. Smoking was not commonly associated with amputation but lowered the mean age at the time of amputation by 5 years (Figure 5).

#### 3.3.4. Mortality

Post-operative in-hospital mortality was 10.2%. A total of 21.8% of the deceased patients were <65 years of age (RR = 2.37, 95% CI 1.85–3.05, *p* < 0.001)**.** Most deceased patients were 65–80 years old (51%) and nearly 1/5th of people >85 years died during admission (Appendix A).

### 3.4. The Questionnaire

Amputations were equally performed across Romania (Figure 2A), while revascularisations were only performed in vascular centres or other university centres (Figure 2B,C). We observed that most amputations occurred in the counties with most revascularisations.

## 4. Discussion

This study evaluates the number and type of amputations performed in Romania between 2016 and 2021 using good quality real-world data from a core pool of 61 hospitals covering approximately half of the cases performed. One similar study has been conducted in Romania but only reported on diabetes-related amputation [18].

This study describes the current situation in the Romanian healthcare system. We identified that PAD, DM, and hypertension are the most common comorbidities found in patients who had an amputation. Furthermore, 5.1% of revascularisations were attempted prior to amputation, suggesting Romanian patients have poor access to specialised vascular surgical care to reduce the number of amputations.

We analysed data from the COVID-19 pandemic and observed that in 2020, the number of all procedures dropped, except for major amputations, which surprisingly stayed the same. Patients were not given adequate options for revascularisation, and the number of such procedures was higher in the pre-pandemic period. We saw that regular specialty doctor visits do not reduce the incidence of major amputations, as outpatient care significantly dropped in 2020 [19]. This may be because the rate of patients with an amputation who are seen by a vascular surgeon is below 20%.

The most likely cause of the ascending trend in revascularisations is probably due to technological advancement in the Romanian healthcare system coupled with the rising number of vascular centres where these procedures are being performed.

### 4.1. Risk of Bias, Missing Data, and Limitations

There is no vascular surgery registry in Romania, and the transition towards digital health data administration is a slow process. Therefore, a risk of bias cannot be excluded. Some of the cases may have been referred to different hospitals (e.g., revascularisation at a university hospital and secondary amputation at a County Hospital), or have procedures in different years (e.g., revascularisation in December and secondary amputation in January). However, the pool of 61 hospitals used for the more detailed analysis covers nearly 45% of the cases reported in 2019, making it a very strong sample.

Aside from this, there are no electronic medical records available about cases treated in private hospitals out of the public financing system (out-of-pocket payment cases) due to the lack of obligation for hospitals to report these cases to NHIH and NSPHMPDB. However, based on expert opinion, these are estimated to represent less than 2% of cases, meaning that the data would have little statistical impact on our conclusions.

Uncertainty due to the data quality of the electronic health records and lack of available data regarding multiple admissions (readmissions) in different hospitals and over a longer period (more than 1 year) can also be seen as a limitation. Moreover, having access only to such a small period of time, the decline in open surgery could not be shown to be statistically significant.

### 4.2. Interpretation and Generalisation

If we compare data from the national database with studies from Western Countries [2,20,21,22], we observe an increase in major amputation rates (mean prevalence in Germany [22], Denmark [2], and Finland [21] was 23.9 per 100,000, compared with 34.2 for Romania). Alternatively, if we compare with our neighbouring country, Hungary [23], the frequency of major amputation is significantly lower (51.3% compared with 34.2% in Romania). The percentage of revascularisation procedures performed prior to amputation can be compared only with Denmark which was 5 times lower than Romania (5.1% compared with 24.8% [2]). However, looking at the raw number of revascularisations, it can be observed that the prevalence of revascularisations in Romania is 64% lower than that in a Western European country (72.4 compared with 202.55) [2,21,24]. Additionally, Romania has a lower percentage of open surgical revascularisations compared to Hungary (59.8% [20] versus 52%), which is still 23.3% higher than Western countries like Denmark [2]. These differences have two possible explanations, which we cannot evaluate because of a missing standardised age format. Differences can be due to the mean age of the total population being higher or lower than the comparison country or due to higher or lower rates of revascularisations.

By looking at administrative comorbidities, we were able to compare only the data from Denmark with our data from Romania where 31.3% [2] of patients with amputation had consulted a vascular surgeon prior to operation, compared to our estimated 19.1%.

With regard to medical comorbidities, our top three comorbidities were hypertension, DM, and PAD. While Denmark has more patients with amputation and HTA (81.2%), in Romania, the highest percentage of those with an amputation coincided with PAD (76.8%). Germany has the highest rate of minor or major amputation rates in patients with DM (85.6% and 63.7%, respectively) [2]. However, the USA has the highest percentage of amputation patients who smoke (14%), closely followed by the United Kingdom (8.9%) [20,21,22,23,24,25]. Appendix A demonstrates the whole analysis in detail.

The percentage of smokers in this type of study is lower than that experienced in clinical practice, suggesting a degree of selection bias. A plausible reason for this discrepancy might be that the data collection and reporting is based on the anamnesis (patient-reported history) rather than a standardised, robust clinical tool to assess smoking status which results in underreporting. A recent study showed that 45.4% of PAD patients who undergo revascularisation also smoke [25].

By comparing the data of the Western European countries (Finland, Denmark, Germany, and the United Kingdom) with the data from this study, it can be deduced that patients in Romania suffer amputation 5.7 years earlier, the rate of revascularisation is 64% lower, and the rate of amputations is 51.7% higher. Due to a lack of data, it cannot be determined if a patient received a vascular consultation prior to an amputation. We were only able to show that 19% of patients had been discharged from a vascular surgery clinic and 17.5% of patients had been operated on by a vascular surgeon. With this, we can only estimate that under 20% of people who received an amputation had been consulted by a vascular surgeon prior to their amputation.

Our data from the 61-hospital cohort showed that only 5.1% of amputated patients had a revascularisation attempt prior to lower extremity amputation. The 30-day amputation mortality with or without prior revascularisation was also high, with a rate of 10.2%.

## 5. Conclusions

The signal from our data indicates that Romanian patients probably have a high risk of amputation, approximately 6 years earlier than Western European countries. However, international data need age standardisation to allow true comparison. PAD is the most prevalent comorbidity for amputation over the age of 40 in Romania (77%), where a high number of cases have both PAD and diabetes (43.8%). In Romania, clinical investigation before lower extremity amputation by a vascular surgeon could reduce the number of amputations. The high number of amputations completed by general surgeons in Romania identifies a necessity to raise greater awareness, not only between patients but also between physicians, about the importance of vascular surgical input on revascularisation decisions. A vascular registry that collects data from all European countries, led by the European Society of Vascular Surgery, would highlight trends and stark international disparity.

## Figures and Tables

**Figure 1 jcm-13-00052-f001:**
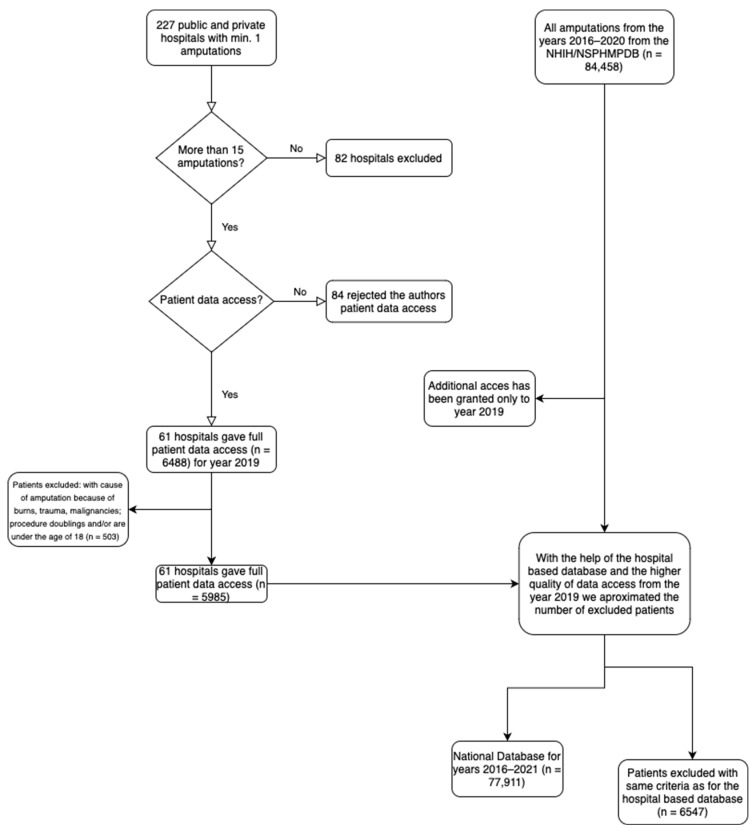
Flow chart used to simplify the explanation for the creation of the hospital-based database and the national database. n indicates the number of patients.

**Figure 2 jcm-13-00052-f002:**
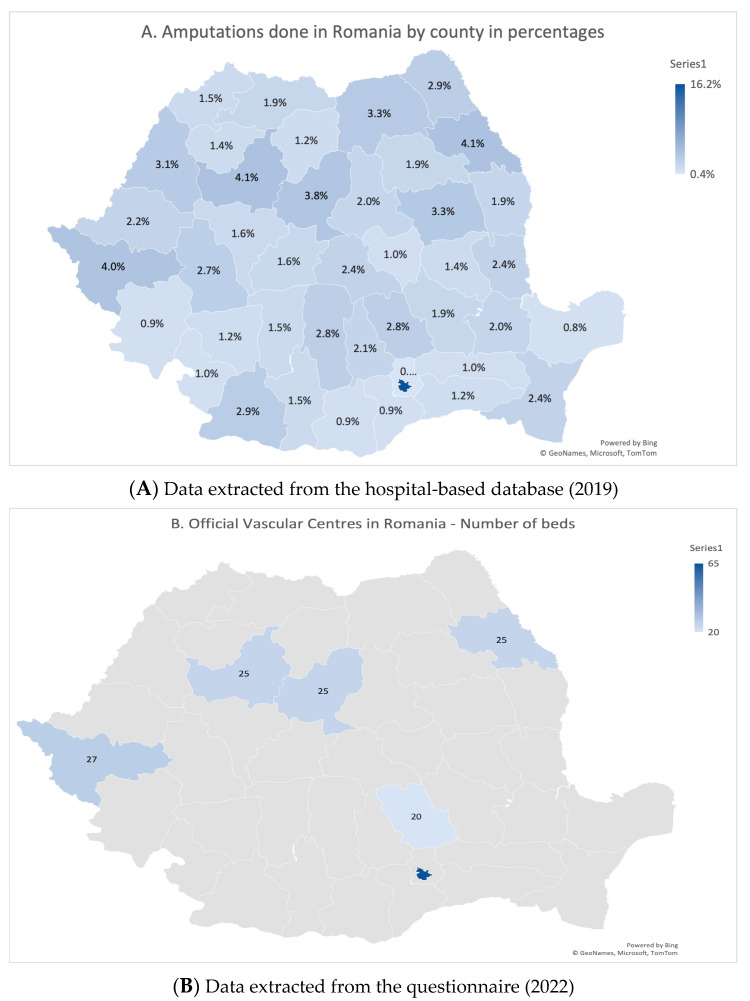
(**A**) Amputation map by county in 2019; (**B**) Number of beds of vascular surgery centers by county in 2022; (**C**) Revascularisations performed by county (in percentages) in 2019.

**Figure 3 jcm-13-00052-f003:**
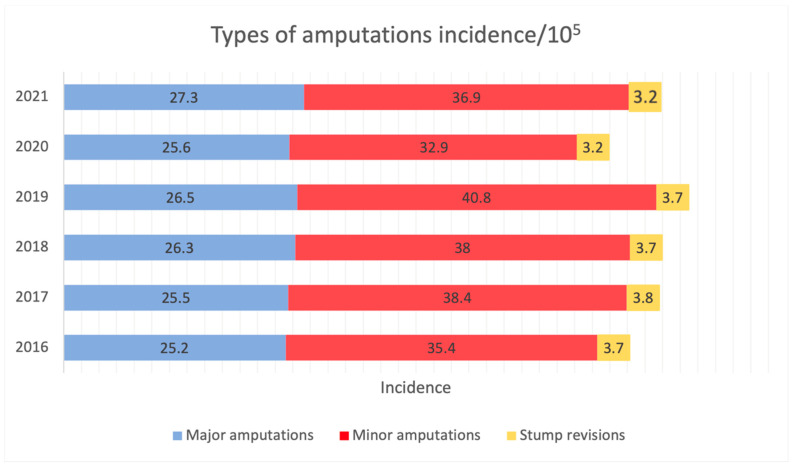
Major amputations, minor amputations, stump revisions, and total amputations incidence/10^5^. Data extracted from the national database (2016–2021).

**Figure 4 jcm-13-00052-f004:**
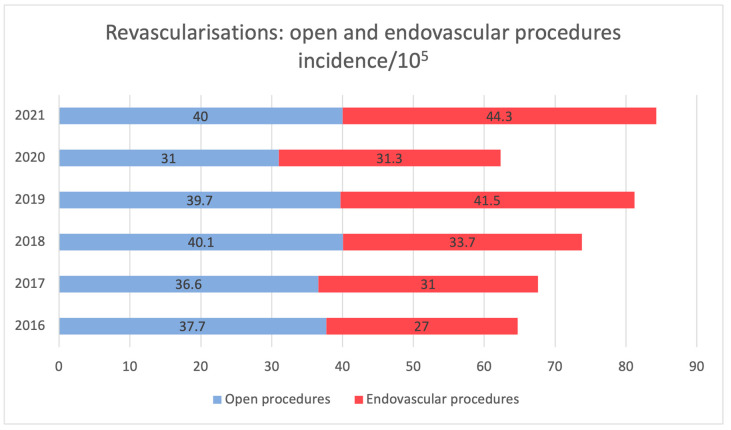
Revascularisations: open and endovascular procedures incidence extracted. Data extracted from the national database (2016–2021).

**Figure 5 jcm-13-00052-f005:**
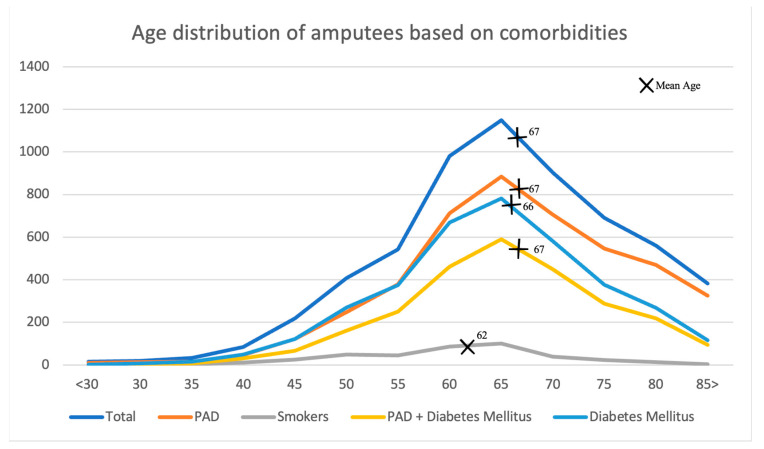
Age distribution of amputees based on comorbidities. Data extracted from the national database (2019).

**Table 1 jcm-13-00052-t001:** Included and excluded cases from the national and hospital-based databases by year and the estimation of cases.

Exclusion Process and Total Number of Included Amputations							
Year	Included Amputations	Excluded Amputations				Total Excluded (*n*%)	Amputations in the Study (*n*%)
		Lesions, Malignancies, Obstructions and Other External Causes (*n*%)	DRG Group for Burns (*n*%)	Age < 18 Years (*n*%)	Doublings (*n*%)		
The national database—Pool of 61 hospitals							
2019	6488	100 (1.5)	15 (0.2)	14 (0.2)	374 (5.8)	503 (7.8)	5985 (92.2)
Hospital-based database—NHIH and/or NSPHMPDB							
2016	13,780 ^a^	212 ^b^	32 ^c^	30 ^d^	794 ^e^	1068 ^f^	12,712 ^g^
2017	14,409 ^a^	222 ^b^	33 ^c^	31 ^d^	831 ^e^	1117 ^f^	13,292 ^g^
2018	14,383 ^a^	222 ^b^	33 ^c^	31 ^d^	829 ^e^	1115 ^f^	13,268 ^g^
2019	14,938 ^a^	230 ^b^	35 ^c^	32 ^d^	861 ^e^	1158 ^f^	13,780 ^g^
2020	12,924 ^a^	199 ^b^	30 ^c^	28 ^d^	745 ^e^	1002 ^f^	11,922 ^g^
2021	14,024 ^a^	216 ^b^	32 ^c^	30 ^d^	808 ^e^	1087 ^f^	12,937 ^g^

Data extracted from the national and hospital-based databases. ^a^ not corrected, ^b^ Estimation—estimated to be 1.5% of the total, ^c^ Estimation—estimated to be 0.2% of the total, ^d^ Estimation—estimated to be 0.2% of the total, ^e^ Estimation—estimated to be 0.2% of the total, ^f^ Estimation—estimated to be 7.8% of the total, ^g^ Estimation—estimated to be 92.2% of the total.

**Table 2 jcm-13-00052-t002:** Comorbidities.

Comorbidities of Cases with/without Amputations				
Comorbidity	With Amputations (*n* = 5985)	Without Amputations (*n* = 3,725,280)	*p*	OR 95% CI
Smoking (*n* = 29,115) No. (% c/% r)	438 (7.3/1.5)	28,677 (0.8/0.2)	<0.001	10.18 (9.2; 11.2)
Hyperlipidemia (*n* = 124,596) No. (% c/% r)	488 (8.2/0.4)	124,108 (3.3/0.2)	<0.001	2.58 (2.4; 2.8)
DM (*n* = 166,616) No. (% c/% r)	3629 (60.6/2.2)	162,987 (4.4/0.1)	<0.001	33.7 (31.9; 35.5)
PAD (*n* = 65 923) No. (% c/% r)	4589 (76.7/7)	61,334 (1.6/0.04)	<0.001	196.37 (184.9; 208.6)
Revascularisation (*n* = 4514) No. (% c/% r)	302 (5/6.7)	4212 (0.1/0.2)	<0.001	46.95 (41.7; 52.9)
DM + Gangrene/ulcer (*n* = 4535) No. (% c/% r)	1936 (32.3/42.7)	2599 (0.1/0.1)	<0.001	684.9 (640.9; 731.9)
PAD and DM (*n* = 234,539) No. (% c/% r)	2624 (43.8/1.1)	231,915 (6.2/0.1)	<0.001	11.76 (11.2; 12.4)
Hypertension (*n* = 415,731) No. (% c/% r)	3199 (53.5/0.8)	412,532 (11.1/0.1)	<0.001	9.22 (8.8; 9.7)
Ischemic Stroke (*n* = 96,857) No. (% c/% r)	646 (10.8/0.7)	91,211 (2.4/0.2)	<0.001	4.86 (4.5; 5.3)
Coronary Disease (*n* = 19,833) No. (% c/% r)	107 (1.8/0.5)	19,726 (0.5/0.2)	<0.001	3.42 (2.8; 4.1)
Carotid Disease (*n* = 11,796) No. (% c/% r)	90 (1.5/0.8)	11,706 (0.3/0.2)	<0.001	4.84 (3.9; 6)
Obesity (*n* = 117,393) No. (% c/% r)	668 (11.2/0.6)	116,725 (3.1/0.2)	<0.001	3.88 (3.6; 4.2)
Alcoholism (*n* = 45,622) No. (% c/% r)	168 (2.8/0.4)	45,454 (1.2/0.2)	<0.001	2.34 (2.0; 2.7)

Data extracted from the national database (2019). No. (% c/% r)—Number (% of the total in a column/% of the total in a row); *n*—number of cases, OR—odds ratio, CI—confidence interval.

## Data Availability

All the data are available on https://docs.google.com/spreadsheets/d/1Wx-xwV2VkVwlWs1fS9AAb6NubKIjzkRmXx6xP_MUfzs/edit?usp=sharing (accesed on 1 October 2023).

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
