# Peer review of "Lower Extremity Amputation and Peripheral Revascularisation Rates in Romania and Their Relationship with Comorbidities and Vascular Care"

_jcm, 2023, doi:10.3390/jcm13010052_

Round 1

Reviewer 1 Report

Comments and Suggestions for Authors

Thank you for the opportunity to review the above article.

Publishing information about the situation in Romania to improve the treatment of peripheral vascular diseases is crucial. The high percentage of amputations can indicate many risk factors in the majority of the population but can also indicate late diagnosis and lack of secondary and tertiary prevention.

Author Response

Dear Reviewer 1,

Thank you for your time reviewing our manuscript and appreciate that you see its merits. We agree this could be the catalyst for important change in Romania

We have attached a revised version as per other reviewers instructions. 

All the best,

Stefan Ionac

Reviewer 2 Report

Comments and Suggestions for Authors

Great effort & job but there are some point need to be considered

what is the implication of this study?

what is the type of research used?

how & who could response for collecting the data and extract?

In paper 

Introduction & Materials and Methods need to be more clear & concise  

what is the reference of definitions? 

results too long & reference should be re-edit & updated 

Comments on the Quality of English Language

need to be improved

Author Response

Dear reviewer 2,

Great effort & job but there are some point need to be considered

We are appreciative of your help in improving our manuscript and have addressed your questions individually below. Where necessary the manuscript has been changed.

  • what is the implication of this study?

This is an incredibly important point and have added text too our introduction to highlight the impact this work could have on amputation reduction across Romania on page 2, lines 71 to 76, redline. This is in addition to our conclusion on page 12, lines 1153 onwards, redline.

  • what is the type of research used?

This study was a retrospective nationwide data-analysis of procedures and outcomes submitted to Romanian national authorities. This is explained and now expanded on Page 2, lines 71 to 76 redline.

  • how & who could response for collecting the data and extract?

Within the methods section on page 2, lines 85 to 88, we explain that This study utilised population-based administrative data collected by these two entities (NHIH and NSPHMDPB) between 2016-2021, to create a singular national level research database.  A patient-level anonymised data set was then created, and only then, extracted data from a group of hospitals, for the year 2019, shared with the study team.” We have amended this as you see to indicate that the anonymised dataset was shared with the study team.

In paper 

  • Introduction & Materials and Methods need to be more clear & concise  

Reviewer 2 has asked for the same change as reviewer 3. Please see our outlined changes explained below in answer to reviewer 3, but in short, we agree and have substantially re-written the methods section.

  • What is the reference of definitions? 

Thank you for asking for this important clarification. References 17 has been added to this section.

  • Results too long & reference should be re-edit & updated 

At thew reviewers request we have edited and updated our references. However, there is a conflict between what reviewer 2 and 3 are asking in regard to our results section. One is asking for less, the other for additions which we hope the reviewer can understand is challenging to address. We have made substantial changes/reductions to our results section which we hope the reviewer appreciates. The manuscript remains within JCM’s allowed word count.

Thank you for your time in helping us improve our manuscript for the benefit of the JCM readership.

Please see attachment for changes.

We thank you for your consideration.

Kind Regards,

Stefan Ionac

Reviewer 3 Report

Comments and Suggestions for Authors

Brief Summary: The authors present findings from a retrospective Romanian study analyzing amputations and revascularization in patients with cardiovascular or diabetic comorbidities from 2016 to 2021. The results include information on the type of surgery, comorbidities, and care organization. The study suggests that Romanian patients may face a higher risk of amputation compared to Western countries.

General Comments: I appreciate the authors' contribution to better understanding the different care pathways across European countries. The study is interesting, but it requires improvement. Specifically, the paper should include more current references, reorganize the Materials and Methods section, check the order of figures, verify figure legends, and present some results in a table for clarity.

Specific Comments:

  1. Introduction:

    • Line 36: Specify the percentage of progress to a critical limb.
    • Lines 38-39: Elaborate on the detailed study cited in '2', clarifying it was conducted only in Denmark from 1997 and indicating the increasing trend of vascular surgeon evaluation.
    • Line 40: Consider whether the citation '3' (data from 2005) is relevant (probably not representative of current practice).
    • Line 41: For reference ‘4’, clarify if stage amputation details (all/minor/major) are available.
  2. Materials and Methods:

    • Figure 1 Flowchart: Improve readability; it is currently too small and unclear.
    • Line 149: Correct the order of figure citations; it seems inconsistent.
    • Inclusion/exclusion: Present inclusion and exclusion criteria for each database when introduced or compile them in a table for clarity.
    • Line 175: Include a reference to define major amputation.
    • Line 184: Cite supplementary table S4.
  3. Results:

    • First section on the national database: If available, present patient characteristics as mentioned in the first sentence of the hospital-based data section.
    • Figure 2: Remove ellipsis (...) in the line related to 2021's data.
    • Supplementary table S6: Place it appropriately (it comes after S7, S8, S9).
    • Data on the national database before and after the COVID pandemic: could be summarize in a table for clarity.
    • Legends in figures 2 and 3: Reverse them for accuracy.
    • Figure 5: Simplify the legend on the graph to display only ‘<30’, ‘30’, ‘35’, ‘40’…
    • Supplementary table S7: Ensure uniformity in the size of numbers.
  4. Discussion:

    • No specific comments

Author Response

Dear Reviewer 3,

You have made some important suggestions which we agree are important. We have addressed these individually below and highlighted where our manuscript has been changed.

Specific Comments:

  1. Introduction:
    • Line 36: Specify the percentage of progress to a critical limb.

We have added this to page 1, lines 36 and 37, redline.

    • Lines 38-39: Elaborate on the detailed study cited in '2', clarifying it was conducted only in Denmark from 1997 and indicating the increasing trend of vascular surgeon evaluation.

This has been corrected on page 1, lines 38 and 39 redline.

    • Line 40: Consider whether the citation '3' (data from 2005) is relevant (probably not representative of current practice).

We agree with the reviewer that this work is old and it may not be representative of current practice across Western Europe and the USA. However, the it is still reflective of the practice in Romania which is why we consider it relevant to this work. We have amended the text to reflect this important viewpoint on page 1, lines 40 to 44, redline.

    • Line 41: For reference ‘4’, clarify if stage amputation details (all/minor/major) are available.

Thank you for this clarification. This has now been added on page 1, line 42, redline.

  1. Materials and Methods:
    • Figure 1 Flowchart: Improve readability; it is currently too small and unclear.

Quite correct. This has now been fixed on page 5, redline.

    • Line 149: Correct the order of figure citations; it seems inconsistent.

Apologies. You should not have needed to request this. We have now amended this mistake.

    • Inclusion/exclusion: Present inclusion and exclusion criteria for each database when introduced or compile them in a table for clarity.

As we have reached the maximum number of allowed tables, we have moved the criteria to the relevant section about each database as the reviewer suggests.

    • Line 175: Include a reference to define major amputation.

Reviewer 2 asked the for same correction. Reference 17 has now been added.

    • Line 184: Cite supplementary table S4.

This has now been added on page 5, line 385, redline, as requested.

  1. Results:
    • First section on the national database: If available, present patient characteristics as mentioned in the first sentence of the hospital-based data section.

We agree with the reviewer that this would be nice data to include but unfortunately it was not available too the study team in the anonymised data extraction. We hope the reviewer understands this is beyond our control.

    • Figure 2: Remove ellipsis (...) in the line related to 2021's data.

Thank you. These have been removed.

    • Supplementary table S6: Place it appropriately (it comes after S7, S8, S9).

This has been now resolved.

    • Data on the national database before and after the COVID pandemic: could be summarize in a table for clarity.

Thank you for asking for this would take us well above the number of allowable tables, as per journal guidelines. This is in addition to multiple other tables being requested by the reviewers. We feel this wouldn’t allow for the necessary commentary to explain the important data and have therefore needed to leave this section as it was to keep within the journal rules.

    • Legends in figures 2 and 3: Reverse them for accuracy.

This has now been done as requested.

    • Figure 5: Simplify the legend on the graph to display only ‘<30’, ‘30’, ‘35’, ‘40’…

Now figure 4. As requested, this has been changed.

    • Supplementary table S7: Ensure uniformity in the size of numbers.

This has been corrected.

  1. Discussion:
    • No specific comments

We are grateful that you consider the discussion as complete and thank you for the time in improving the overall content.

We thank you for your consideration.

Kind Regards,

Stefan Ionac

Round 2

Reviewer 3 Report

Comments and Suggestions for Authors

thank you for the different modifications which clearly improve the manuscript